# Premature Vascular Aging with Features of Plaque Vulnerability in an Atheroprone Mouse Model of Hutchinson–Gilford Progeria Syndrome with *Ldlr* Deficiency

**DOI:** 10.3390/cells9102252

**Published:** 2020-10-08

**Authors:** Rosa M. Nevado, Magda R. Hamczyk, Pilar Gonzalo, María Jesús Andrés-Manzano, Vicente Andrés

**Affiliations:** 1Centro Nacional de Investigaciones Cardiovasculares Carlos III (CNIC), 28029 Madrid, Spain; rosamaria.nevado@cnic.es (R.M.N.); pgonzalo@cnic.es (P.G.); mjandres@cnic.es (M.J.A.-M.); 2Centro de Investigación Biomédica en Red de Enfermedades Cardiovasculares (CIBERCV), 28029 Madrid, Spain; 3Departamento de Bioquímica y Biología Molecular, Instituto Universitario de Oncología (IUOPA), Universidad de Oviedo, 33006 Oviedo, Spain; hamczykmagda.uo@uniovi.es

**Keywords:** aging, atherosclerosis, lamin A, progeria, vulnerable plaque

## Abstract

Hutchinson–Gilford progeria syndrome (HGPS) is among the most devastating of the laminopathies, rare genetic diseases caused by mutations in genes encoding nuclear lamina proteins. HGPS patients age prematurely and die in adolescence, typically of atherosclerosis-associated complications. The mechanisms of HGPS-related atherosclerosis are not fully understood due to the scarcity of patient-derived samples and the availability of only one atheroprone mouse model of the disease. Here, we generated a new atherosusceptible model of HGPS by crossing progeroid *Lmna^G609G/G609G^* mice, which carry a disease-causing mutation in the *Lmna* gene, with *Ldlr*^−/−^ mice, a commonly used preclinical atherosclerosis model. *Ldlr*^−/−^*Lmna^G609G/G609G^* mice aged prematurely and had reduced body weight and survival. Compared with control mice, *Ldlr*^−/−^*Lmna^G609G/G609G^* mouse aortas showed a higher atherosclerosis burden and structural abnormalities typical of HGPS patients, including vascular smooth muscle cell depletion in the media, adventitial thickening, and elastin structure alterations. Atheromas of *Ldlr*^−/−^*Lmna^G609G/G609G^* mice had features of unstable plaques, including the presence of erythrocytes and iron deposits and reduced smooth muscle cell and collagen content. *Ldlr*^−/−^*Lmna^G609G/G609G^* mice faithfully recapitulate vascular features found in patients and thus provide a new tool for studying the mechanisms of HGPS-related atherosclerosis and for testing therapies.

## 1. Introduction

Lamins are key components of the nuclear envelope that play pivotal roles in maintaining a proper nuclear structure and function. Mutations in genes encoding lamins lead to a variety of diseases collectively termed laminopathies, which affect one or multiple tissues, including muscular, cardiac, adipose, and osseous tissue. One of the more severe laminopathies is Hutchinson–Gilford progeria syndrome (HGPS), which is caused by progerin, an anomalous variant of lamin A. Progerin arises via aberrant splicing of *LMNA* mRNA, typically due to a de novo c.1824C>T;p.G608G mutation [1,2]. Very early in life, HGPS patients begin to show symptoms characteristic of advanced age, including alopecia, loss of subcutaneous fat, skin wrinkling, bone and joint problems, and cardiovascular disease [3,4]. Most HGPS patients die from atherosclerosis-related causes, mainly myocardial infarction or stroke, at an average age of 14.6 years [5,6]. Autopsies reveal that apart from atherosclerotic lesions, HGPS arteries have severe structural abnormalities, including excessive calcification, adventitial thickening, smooth muscle cell loss and extracellular matrix accumulation in the media, and elastin structure alterations [5,7,8].

HGPS is an extremely rare disorder (prevalence about 1 in 18 million) [9], and therefore, human cardiovascular tissue samples are scarce and limited to post-mortem specimens. There is therefore a need for animal models for the study of disease mechanisms, especially disease initiation and progression, in order to pave the way for effective treatments. Several progerin-expressing mouse models have been generated [10]; however, they do not develop atherosclerosis, the cause of death in HGPS. This is because mice, unlike humans, are resistant to atherosclerotic disease due to an atheroprotective lipid profile featuring high circulating levels of high-density lipoprotein (HDL) and low levels of low-density lipoprotein (LDL) [11]. Conducting atherosclerosis research with mice therefore requires genetic alteration of their lipid profile, often in combination with high-fat, high-cholesterol feeding. The most-widely used atherosclerosis mouse models are *Apoe*^−/−^ and *Ldlr*^−/−^ mice, which lack apolipoprotein E and the LDL receptor, respectively. However, studies with these models do not always lead to the same conclusions, mainly because the *Apoe* and *Ldlr* genes have different expression patterns in tissues and their deficiency yields different lipid profiles, resulting in atherosclerotic lesions with slightly different features and topology [12]. Thus, atherosclerotic plaques in *Apoe*^−/−^ mice are more similar to those observed in humans, but the circulating lipid profile of *Ldlr*^−/−^ mice is more similar to that of hypercholesterolemic humans. It is therefore preferable to use both models in order to avoid the reporting of effects specific only to *Apoe* or *Ldlr* deficiency and thus identify effects genuinely caused by progerin expression.

We recently generated an *Apoe*^−/−^*Lmna^G609G/G609G^* mouse model that resembles most features of vascular disease found in HGPS patients, including accelerated atherosclerosis [13]. To validate and extend the results obtained with the *Apoe*^−/−^*Lmna^G609G/G609G^* model, we have now generated and characterized atherosclerosis development in progerin-expressing *Ldlr*^−/−^*Lmna^G609G/G609G^* mice. This new mouse model displays accelerated organismal and vascular aging and has a phenotype similar but not identical to that of *Apoe*^−/−^*Lmna^G609G/G609G^* mice. Comparison of the two models allowed us to identify key features of atherosclerosis associated with HGPS.

## 2. Materials and Methods

### 2.1. Study Approval

Animal experimental procedures followed EU Directive 2010/63EU and Recommendation 2007/526/EC, enforced in Spanish law under Real Decreto 53/2013. Animal protocols were approved by the local ethics committees and the Animal Protection Area of the Comunidad Autónoma de Madrid (PROEX 149.0/20).

### 2.2. Mice

The *Ldlr*^−/−^*Lmna^G609G/G609G^* mouse line was generated by crossing *Ldlr*^−/−^ mice (B6.129S7-*Ldlr^tm1Her^*/J, stock no. 002207, The Jackson Laboratory, Bar Harbor, ME, USA) with *Lmna^G609G/+^* mice [14]. We used male and female *Ldlr*^−/−^*Lmna^G609G/G609G^*, *Ldlr*^−/−^*Lmna*^*G609G*/+^, and *Ldlr*^−/−^*Lmna*^+/+^ mice on a C57BL/6J genetic background.

Mice were housed in a specific pathogen-free facility in individually ventilated cages with 12 h light/12 h dark cycle at a temperature of 22 ± 2 °C, 50% relative humidity (range 45–60%). Mice had ad libitum access to water and food (normal chow diet: D184, SAFE, Augy, France, and Rod18-A, LASQCdiet, Soest, Germany). When indicated, eight-week-old mice were placed for eight weeks on a high-fat diet (HFD) containing 10.7% total fat and 0.75% cholesterol (S9167-E011, Ssniff, Soest, Germany). Except for longevity studies, animals were sacrificed by CO_2_ inhalation at 16 weeks of age after overnight fasting.

### 2.3. Longevity Studies

Beginning at four weeks of age, mice were weighed and inspected for health and survival at least once a week. Animals that met humane end-point criteria were euthanized and the deaths recorded. We excluded from the analysis mice euthanized due to causes unrelated to phenotype, e.g., intermale aggression.

### 2.4. Hematology and Serum Biochemical Analysis

Mice were fasted overnight for all blood analyses. For hematology, blood samples were collected in Microvette 100 EDTA tubes (20.1278, Sarstedt, Nümbrech, Germany) and analyzed with a PENTRA 80 hematology analyzer (Horiba, Kyoto, Japan). For lipid profile analysis, blood samples were collected in plastic tubes, incubated at room temperature (RT) to allow clotting, and centrifuged at 1900× *g* for 10 min (4 °C). Collected serum was centrifuged for 10 min at maximum speed (4 °C), harvested, and stored at −80 °C. When the serum sample volume was insufficient for the analysis, specimens from 2–3 mice of the same genotype were pooled. Hemolyzed specimens were excluded from testing. The serum content of total cholesterol, free cholesterol, HDL, and LDL was measured using a Dimension RxL Max Integrated Chemistry System (Siemens Healthineers, Erlangen, Germany). All analyses were performed by specialized staff at the CNIC Animal Facility.

### 2.5. Oil Red O Staining and Quantification of Atherosclerosis Burden

Aortic atherosclerosis burden was quantified as previously described [13]. Briefly, mouse aortas were fixed with 4% formaldehyde in phosphate-buffered saline (PBS), cleaned of fatty tissue, and stained with 0.2% Oil Red O (ORO, O0625, Sigma, St. Louis, MO, USA). The aorta was opened longitudinally and pinned flat for planimetric analysis. Images were acquired with a digital camera (UC30, OLYMPUS, Tokyo, Japan) mounted on a stereo microscope (SZX3, OLYMPUS). Lesion area was quantified as the percentage of ORO-stained aortic surface by a researcher blinded to genotype using SigmaScan Pro 5 software (Systat Software Inc., San Jose, CA, USA).

### 2.6. Histology

The mouse aortic arch and aortic root were fixed with 4% formaldehyde in PBS, dehydrated to xylene, and embedded in paraffin. Tissue was cut in 4 μm sections using a microtome (RM2245, Leica, Wetzlar, Germany), and specimens were stained with hematoxylin–eosin, Masson trichrome, Van Gieson, or Perls Prussian blue. Stained sections were scanned with a NanoZoomer-RS scanner (Hamamatsu Photonics K.K., Hamamatsu, Japan). Plaque area and the aortic perimeter affected by atherosclerosis were analyzed with NDP.view2 (Hamamatsu Photonics K.K.) using Freehand Region and Freehand Line annotation tools. Atheroma collagen content, adventitia thickness, and media thickness were quantified using ImageJ Fiji software [15]. Collagen content in atheroma was assessed in Masson trichrome-stained aortic root sections by image color deconvolution and quantification of the percentage of blue area in plaque regions that were drawn manually. Adventitia and media thickness were assessed in Van Gieson-stained aortic arch sections by drawing two circular band regions (one for the media and one for the adventitia). These regions of interest were linearized, and the mean thickness of each aortic layer was measured using a specific macro. Then the adventitia-to-media thickness ratio was calculated. Aorta and atheroma plaque features were analyzed by a researcher blinded to genotype in approximately three sections per animal, and mean values were used for statistical analysis.

### 2.7. Fluorescent Immunohistochemistry

The mouse aortic arch and aortic root were fixed with 4% formaldehyde in PBS, dehydrated to xylene, embedded in paraffin, and cut into 4 μm sections using a microtome (RM2245, Leica). Antigens were retrieved with 10 mM sodium citrate buffer (pH 6), and samples were blocked for 1 h at room temperature (RT) with 5% bovine serum albumin (BSA) and 5% normal goat serum in PBS. Sections were incubated for 2 h at RT with an anti-α-smooth muscle actin (SMA)-Cy3 antibody (C6198, Sigma, 1:200) and Hoechst 33342 stain (B2261, Sigma) diluted in 2.5% normal goat serum in PBS. Sections were mounted in Fluoromount G imaging medium (00-4958-02, Affymetrix eBioscience, Santa Clara, CA, USA). Fluorescence images were acquired with a LSM 700 confocal microscope (Zeiss, Oberkochen, Germany) and analyzed using ImageJ Fiji software [15] by a researcher blinded to genotype. SMA content was quantified in the red channel by measuring the percentage of red area in the region of interest. Nuclei count was assessed in the blue channel by quantifying the number of nuclei per area in the region of interest. Regions of interests were drawn manually and included: aortic media for aortic arch sections, and atheroma, atheroma without necrotic core, and fibrous cap in the atheroma (6 µm band below the endothelium) for aortic root sections. Aorta and atheroma plaque features were analyzed in approximately three sections per animal and mean values used for statistical analysis.

### 2.8. Statistical Analysis

Data distribution was assessed with Kolmogorov–Smirnov and D’Agostino–Pearson normality tests. For parametric data, a two-tailed *t*-test was used, and Welch’s correction was applied if the groups had unequal variances. A two-tailed Mann–Whitney test was used for nonparametric data. Repeated measures data (body weight curves) were analyzed by fitting a mixed effects model with the Geisser–Greenhouse correction. Kaplan–Meier survival curves were compared by log-rank (Mantel–Cox) test. Experimental data are presented as the mean for parametric data (error bars indicate SEM) or the median for nonparametric data. Statistical analysis was performed with Prism 5 and 8 (GraphPad, San Diego, CA, USA). Differences were considered significant at *p* < 0.05.

## 3. Results

To generate a new mouse model to study atherosclerosis in HGPS, we crossed atheroprone *Ldlr*^−/−^ mice with progeroid *Lmna^G609G^* mice, which carry a c.1827C>T; p.G609G mutation in the *Lmna* gene and, like HGPS patients, produce progerin through aberrant splicing [14,16]. Homozygous *Ldlr*^−/−^*Lmna^G609G/G609G^* and heterozygous *Ldlr*^−/−^*Lmna*^*G609G*/+^ mice both had a shorter lifespan and lower body weight than control *Ldlr*^−/−^*Lmna*^+/+^ mice with normal lamin A expression (Figure 1a,b for males; Figure A1a,b for females) and showed an aging phenotype with progressive fat loss closely resembling that of progeric mice without *Ldlr* deficiency. HGPS patients are heterozygous for the *LMNA* mutation; however, *Lmna^G609G^* mice seem to be more resistant to progerin-induced damage [14], and all further studies were performed with homozygous *Ldlr*^−/−^*Lmna^G609G/G609G^* mice because their phenotype is more severe and more similar to that seen in human patients.

To induce atherosclerosis, mice were fed a HFD for eight weeks and sacrificed at 16 weeks of age. *Ldlr*^−/−^*Lmna*^+/+^ and *Ldlr*^−/−^*Lmna^G609G/G609G^* mice had similar post-HFD serum levels of total cholesterol, free cholesterol, LDL, and HDL (Appendix A). Nevertheless, planimetric analysis of ORO-stained aortas revealed a higher atherosclerosis burden in the aortic arch and thoracic aorta of *Ldlr*^−/−^*Lmna^G609G/G609G^* mice (Figure 2a for males; Figure A2a for females). Although *Ldlr*^−/−^ mice typically do not develop substantial atherosclerosis on a low-fat diet [12], we observed modestly but significantly elevated lesion formation in the thoracic aorta of 16-week-old normal chow-fed *Ldlr*^−/−^*Lmna^G609G/G609G^* mice (Appendix A), and this was independent of serum cholesterol level (Appendix A). We next characterized the extent of atherosclerosis in transverse aortic root sections from HFD-fed animals. Although *Ldlr*^−/−^*Lmna^G609G/G609G^* mice had a lower plaque area than *Ldlr*^−/−^*Lmna*^+/+^ controls, they had a higher percentage of the aortic perimeter affected by atherosclerosis, an effect that was more pronounced in males and in aortic root sections more distal from the heart (Figure 2b for males; Figure A2b for females and Appendix A). Most *Ldlr*^−/−^*Lmna^G609G/G609G^* animals showed only modest inflammation and fibrosis in the cardiac tissue adjacent to the aortic root (Figure 2b for males; Figure A2b for females).

Disease severity is defined not only by atherosclerosis burden but also by lesion composition, which also predicts adverse outcomes. We therefore analyzed lesion characteristics, especially those associated with a vulnerable plaque phenotype. Fluorescence immunohistochemistry against SMA showed a lower atheroma content of smooth muscle cells in HFD-fed *Ldlr*^−/−^*Lmna^G609G/G609G^* mice than in *Ldlr*^−/−^*Lmna*^+/+^ controls (Figure 3a for males; Figure A3a for females). Smooth muscle content was also lower in the fibrous cap of *Ldlr*^−/−^*Lmna^G609G/G609G^* mouse atheromas (Figure 3a for males; Figure A3a for females), and this was accompanied by a lower collagen content in lesions, revealed by Masson trichrome staining (Figure 3b for males; Figure A3b for females).

Histological analysis showed a substantial presence of erythrocytes even in small plaques in fat-fed *Ldlr*^−/−^*Lmna^G609G/G609G^* mice but not in control mice, suggesting plaque disruption (Figure 4a for males; Figure A4a for females). Accordingly, Perls Prussian blue staining revealed iron deposits in atheromas of some mutant mice (Figure 4b for males; Figure A4b for females), an indication of hemorrhage and erythrocyte phagocytosis by macrophages. We also found evidence of thrombus formation in *Ldlr*^−/−^*Lmna^G609G/G609G^* mice (Appendix A).

Because autopsies of HGPS patients demonstrated pathological changes in the arterial wall [5,7,8], we next analyzed the structure of the aorta and coronary arteries in HFD-fed animals. Compared with *Ldlr^−/−^Lmna^+/+^* controls, aortas of *Ldlr*^−/−^*Lmna^G609G/G609G^* mice displayed medial and adventitial thickening (Figure 5a for males; Figure A5a for females). The adventitia-to-media ratio was higher in *Ldlr*^−/−^*Lmna^G609G/G609G^* males, but the difference did not reach statistical significance in females (Figure 5a for males; Figure A5a for females). The aortas of mutant mice showed severe depletion of smooth muscle cells in the medial layer, as assessed by both SMA content and nuclei count (Figure 5b for males; Figure A5b for females). Loss of smooth muscle cells was accompanied by decreased waviness of the elastin layers and increased extracellular matrix deposition (Figure 5 for males; Figure A5 for females). We also found adventitial fibrosis and smooth muscle cell loss in the coronary arteries of *Ldlr*^−/−^*Lmna^G609G/G609G^* mice (Appendix A). Similar to HGPS patients, progerin-expressing mice showed aortic valve defects, including increased fibrosis and reduced cellularity in the valve leaflets (Appendix A). In addition, HFD-fed *Ldlr*^−/−^*Lmna^G609G/G609G^* mice showed some hematological alterations that varied slightly between males and females (Appendix A).

## 4. Discussion

In this study, we describe a new atheroprone model of HGPS, a very rare premature aging syndrome resulting in death from atherosclerosis complications. Post-mortem analysis of very low numbers of patients has identified end-stage cardiovascular pathologies; however, the study of disease mechanisms requires in vitro and in vivo models of HGPS. Several animal models of HGPS have been reported, including mice with systemic and tissue-restricted progerin expression [13,14,17,18,19,20,21] and a pig HGPS model [22]. Most models, at least to some extent, exhibit structural abnormalities in the arterial wall; however, progeriod animal models without additional genetic modifications to alter lipid metabolism lacked atherosclerotic disease, indicating that progerin expression in the absence of pro-atherogenic conditions is not sufficient to induce atheroma plaque formation.

We recently generated *Apoe*^−/−^*Lmna^G609G/G609G^* mice, the first animal model to show progerin-induced acceleration of atherosclerosis [13,23,24]. Investigation of HGPS disease mechanisms with this model revealed endoplasmic reticulum stress and the unfolded protein response as possible pathways underlying increased atherosclerosis and smooth muscle cell death in the aortic wall [25]. Since some of the observed alterations might be caused, at least in part, by the lack of apolipoprotein E, we sought to generate an alternative mouse model of progerin-dependent atherosclerosis development to validate and extend the findings in *Apoe*^−/−^*Lmna^G609G/G609G^* mice. We report here the generation and characterization of *Ldlr*^−/−^*Lmna^G609G/G609G^* mice, the second atherosusceptible model for the study of HGPS-related atherosclerosis and for the assessment of novel therapies. The *Ldlr*- and *Apoe*-deficient HGPS models had postnatal growth defects and lifespan shortening similar to atheroresistant *Lmna^G609G/G609G^* mice (median survival in males, 19.9 weeks for *Ldlr*^−/−^*Lmna^G609G/G609G^* mice and 18.15 weeks for *Apoe*^−/−^*Lmna^G609G/G609G^* mice; survival in female *Apoe*^−/−^*Lmna^G609G/G609G^* mice was not reported). Moreover, both *Ldlr*^−/−^*Lmna^G609G/G609G^* and *Apoe*^−/−^*Lmna^G609G/G609G^* models showed key hallmarks of HGPS-associated vascular disease, including augmented atherosclerosis, smooth muscle cell loss in the media, and adventitial thickening. A more detailed comparison of vascular features in both models revealed similar but not identical phenotypes (Table B1, only male data are compared because data on female *Apoe*^−/−^*Lmna^G609G/G609G^* mice are unavailable). For instance, *Ldlr*^−/−^*Lmna^G609G/G609G^* mice had less pronounced adventitial thickening and less fibrosis and inflammation in tissue neighboring the aortic root than *Apoe*^−/−^*Lmna^G609G/G609G^* mice. Furthermore, plaque area did not increase in the aortic root of *Ldlr*^−/−^*Lmna^G609G/G609G^* mice, which may be explained by the fact that *Ldlr*-deficiency yields lower cholesterol levels than *Apoe*-deficiency [12]. On the other hand, *Ldlr*^−/−^*Lmna^G609G/G609G^* mice showed higher atherosclerosis burden in the thoracic aorta and a greater reduction in smooth muscle content in plaques than *Apoe*^−/−^*Lmna^G609G/G609G^* mice. The atherosclerosis assessment in different aortic regions of both atheroprone progeria models indicates that the key features of progerin-induced atherosclerosis are its ubiquity (a very high fraction of the aortic surface affected) and its highly unstable phenotype (Table B1). Based on these findings, we conclude that exacerbated atherosclerosis and its complications in *Apoe*^−/−^*Lmna^G609G/G609G^* and *Ldlr*^−/−^*Lmna^G609G/G609G^* mice are bona fide effects of progerin expression, independent of the genetic manipulation required to establish an atherosusceptible environment in preclinical mouse models. We consider both atheroprone homozygous progeria models adequate for studying mechanisms of progerin-induced vascular aging, especially atherosclerosis. As reported before [26,27,28], calcification is the only aspect of vascular aging that is better approached by using heterozygous *Lmna^G609G/+^* mice, which live longer and therefore exhibit substantial calcium-phosphate deposition in their arteries and valves.

Integration of the data from *Ldlr*^−/−^*Lmna^G609G/G609G^* and *Apoe*^−/−^*Lmna^G609G/G609G^* mouse models and comparison with progerin-independent atherosclerosis suggest a model of progerin-triggered atherosclerosis progression (Figure 6). During progerin-independent atherosclerosis, lesions form at arterial sites with disturbed blood flow, which activates endothelial cells and triggers uptake of LDL particles and the recruitment of blood-borne immune cells to the inflamed vessel wall. As the plaque grows, smooth muscle cells move from the media to the intima, where they proliferate and produce collagen to form a fibrous cap that protects from plaque rupture (Figure 6). During progerin-driven atherosclerosis, the pro-atherogenic stimulus originates within the artery wall (direct or indirect effects of progerin-induced smooth muscle cell death [13,23,25]) and affects almost the entire vessel surface. Atherosclerotic lesions therefore form throughout the intima, unlike plaques in normal atherosclerosis, which typically grow eccentrically. Nevertheless, plaque growth in the presence of progerin, at least in large vessels, may be limited by plaque vulnerability, since the fibrous cap is absent or thin and disorganized in HGPS atheromas, making plaques more prone to rupture and the formation of a life-threatening thrombus (Figure 6).

Atherosclerosis is a major contributor to adverse events and deaths from cardiovascular disease, and its onset and clinical manifestation are highly variable between normally-aging individuals [29]. Studying diseases that drastically accelerate atherogenesis in the absence of traditional cardiovascular risk factors, such as HGPS, may therefore help to identify unknown factors that speed up this process [30]. Moreover, low amounts of progerin have been detected in cells and arteries of elderly individuals [5,31], suggesting that progerin may also play a role in physiological aging and HGPS-unrelated atherosclerosis and its life-threatening complications [10,30].

## Figures and Tables

**Figure 1 cells-09-02252-f001:**
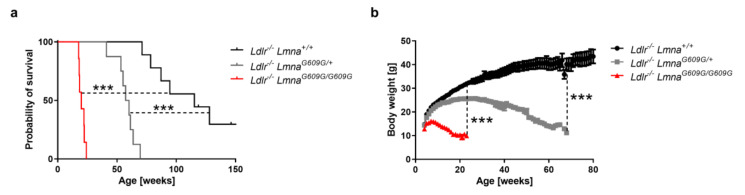
Shortened survival and reduced body weight in male *Ldlr*^−/−^*Lmna^G609G/G609G^* and *Ldlr^−/−^Lmna^G609G/+^* mice. (**a**) Kaplan–Meier survival curves of *Ldlr^−/−^Lmna^+/+^* mice (*n* = 9), *Ldlr^−/−^Lmna^G609G/+^* mice (*n* = 8), and *Ldlr*^−/−^*Lmna^G609G/G609G^* mice (*n* = 7) fed normal chow. (**b**) Body weight curves of *Ldlr^−/−^Lmna^+/+^* mice (*n* = 9), *Ldlr^−/−^Lmna^G609G/+^* mice (*n* = 8), and *Ldlr*^−/−^*Lmna^G609G/G609G^* mice (*n* = 7) fed normal chow. Body weight is shown until death for *Ldlr^−/−^Lmna^G609G/+^* and *Ldlr*^−/−^*Lmna^G609G/G609G^* mice and until 80 weeks of age for *Ldlr^−/−^Lmna^+/+^* mice. Data are presented as mean ± SEM in (**b**). Statistical analysis was performed by log-rank test in (**a**) and by mixed effects model with the Geisser–Greenhouse correction in (**b**). ***, *p* < 0.001.

**Figure 2 cells-09-02252-f002:**
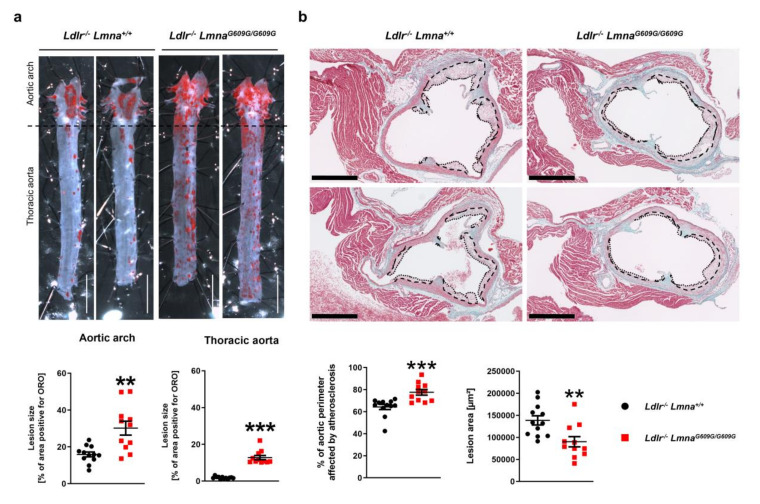
Male fat-fed *Ldlr*^−/−^*Lmna^G609G/G609G^* mice have an elevated atherosclerosis burden in the aortic arch and thoracic aorta and a higher percentage of the aortic perimeter affected by atherosclerosis in the aortic root. Male mice were fed a high-fat diet from eight weeks of age and were sacrificed at 16 weeks of age. (**a**) Representative images of aortas stained with Oil Red O (ORO) and quantification of atherosclerosis burden (the percentage of the aortic surface positive for ORO) in the aortic arch and thoracic aorta of *Ldlr^−/−^Lmna^+/+^* mice (*n* = 12) and *Ldlr*^−/−^*Lmna^G609G/G609G^* mice (*n* = 11). Scale bar, 5 mm. (**b**) Representative images of aortic root sections stained with Masson trichrome and quantification of plaque area and percentage of the aortic perimeter affected by atherosclerosis in *Ldlr^−/−^Lmna^+/+^* mice (*n* = 12) and *Ldlr*^−/−^*Lmna^G609G/G609G^* mice (*n* = 11). Each point represents the mean for three aortic root regions per mouse (see Appendix A for data analyzed for each region separately). Dashed lines indicate the medial perimeter (the last layer of elastin), and dotted lines indicate the luminal surface of the atheroma plaque. Scale bar, 500 µm. Data are presented as mean ± SEM. Statistical analysis was performed by two-tailed *t*-test with Welch’s correction in (**a**) and by two-tailed *t*-test in (**b**). **, *p* < 0.01; ***, *p* < 0.001.

**Figure 3 cells-09-02252-f003:**
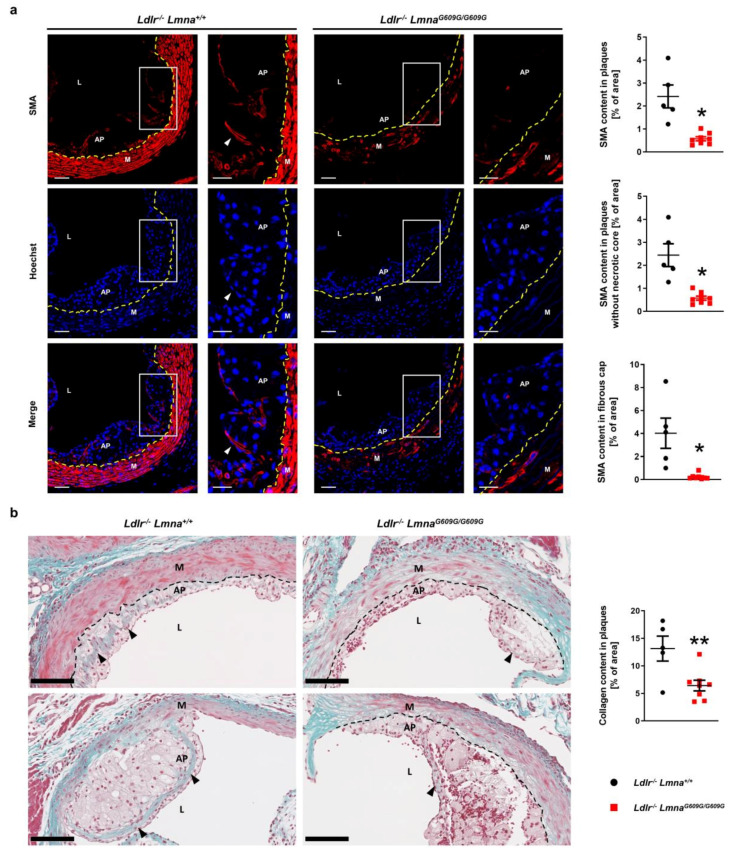
Lower smooth muscle cell and collagen content in atheromas of fat-fed male *Ldlr*^−/−^*Lmna^G609G/G609G^* mice. Male mice were fed a high-fat diet from eight weeks of age and were sacrificed at 16 weeks of age. (**a**) Representative fluorescence images of aortic root sections stained with an anti-α-smooth muscle actin (SMA) antibody (red) and Hoechst 33342 nuclear dye (blue). Discontinuous yellow lines indicate the separation between the media and atheroma plaques. White arrowheads indicate smooth muscle cells in the fibrous cap. Scale bar, 50 µm (non-magnified images) and 25 µm (magnified images). Graphs show smooth muscle content in atheromas (% of area positive for SMA) (top graph), in the cellular part of atheromas (without necrotic cores; middle graph), and in the fibrous cap (6 µm band below the endothelium; bottom graph) of *Ldlr^−/−^Lmna^+/+^* mice (*n* = 5) and *Ldlr*^−/−^*Lmna^G609G/G609G^* mice (*n* = 8). (**b**) Representative images of aortic root sections stained with Masson trichrome. Discontinuous black lines indicate the separation between the media and atheroma plaques. Black arrowheads indicate collagen. Scale bar, 100 µm. The graph shows collagen content (% of blue area) in atheroma plaques of *Ldlr^−/−^Lmna^+/+^* mice (*n* = 5) and *Ldlr*^−/−^*Lmna^G609G/G609G^* mice (*n* = 8). Data are presented as mean ± SEM. Each point represents the mean for three aortic root sections per mouse. Statistical analysis was performed by two-tailed *t*-test with Welch’s correction in (**a**) and by two-tailed *t*-test in (**b**). *, *p* < 0.05; **, *p* < 0.01. AP, atheroma plaque; M, media; L, lumen.

**Figure 4 cells-09-02252-f004:**
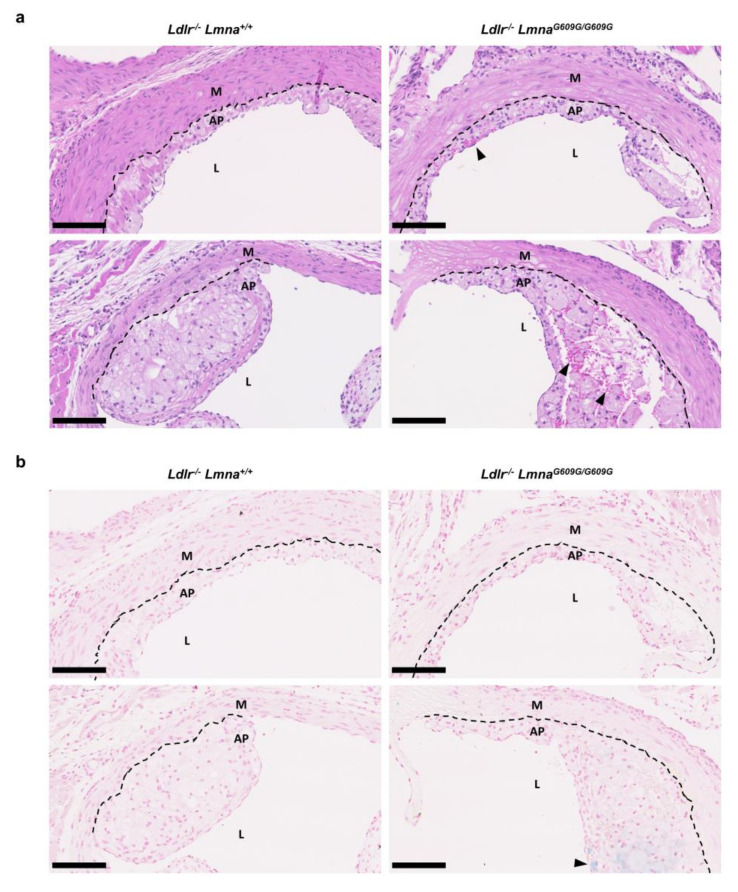
Presence of erythrocytes and iron deposits in atheromas of fat-fed male *Ldlr*^−/−^*Lmna^G609G/G609G^* mice. Male mice were fed a high-fat diet from eight weeks of age and sacrificed at 16 weeks of age. (**a**) Representative images of aortic root sections of *Ldlr^−/−^Lmna^+/+^* and *Ldlr*^−/−^*Lmna^G609G/G609G^* mice stained with hematoxylin–eosin. Black arrowheads indicate erythrocytes. Scale bar, 100 µm. (**b**) Representative images of aortic root sections of *Ldlr^−/−^Lmna^+/+^* and *Ldlr*^−/−^*Lmna^G609G/G609G^* mice stained with Perls Prussian blue. Black arrowheads indicate iron deposits. Scale bar, 100 µm. Discontinuous black lines mark the separation between the media and atheroma plaques. AP, atheroma plaque; M, media; L, lumen.

**Figure 5 cells-09-02252-f005:**
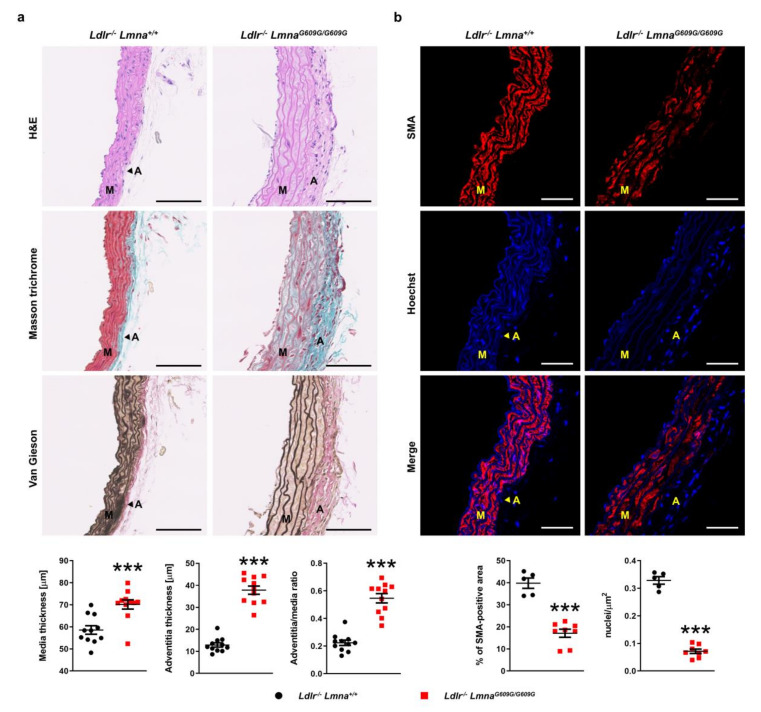
Pathological changes in the aortic wall of fat-fed male *Ldlr*^−/−^*Lmna^G609G/G609G^* mice. Male mice were fed a high-fat diet from eight weeks of age and sacrificed at 16 weeks of age. (**a**) Representative images of aortic arch sections stained with hematoxylin–eosin (H&E), Masson trichrome, and Van Gieson stain. Scale bar, 100 µm. Graphs show medial and adventitial thickness and the adventitia-to-media ratio in *Ldlr^−/−^Lmna^+/+^* mice (*n* = 11) and *Ldlr*^−/−^*Lmna^G609G/G609G^* mice (*n* = 11). (**b**) Representative fluorescence images of aortic arch sections stained with an anti-α-smooth muscle actin (SMA) antibody (red) and Hoechst 33342 nuclear dye (blue). Scale bar, 50 µm. Graphs show the percentage of SMA-positive area and nuclei count/area in the media of *Ldlr^−/−^Lmna^+/+^* mice (*n* = 5) and *Ldlr*^−/−^*Lmna^G609G/G609G^* mice (*n* = 8). Data are presented as mean ± SEM. Each point represents the mean for three aortic arch sections per mouse. Statistical analysis was performed by two-tailed *t*-test with Welch’s correction in (**a**) and by two-tailed *t*-test in (**b**). ***, *p* < 0.001. M, media; A, adventitia.

**Figure 6 cells-09-02252-f006:**
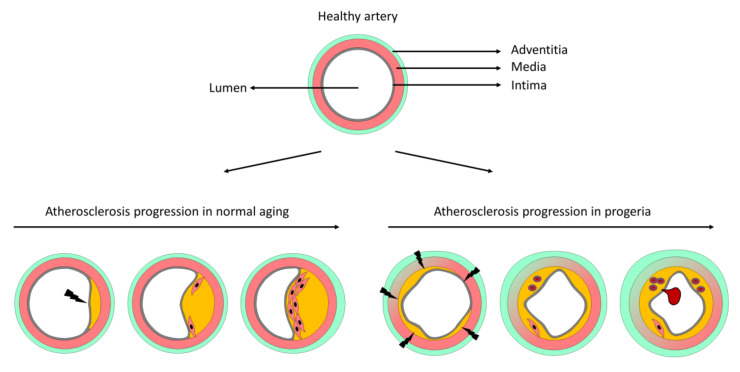
Proposed model of atherosclerosis progression in normal versus progerin-induced mouse aging. A healthy artery is composed of three layers, the intima (containing the endothelial cell monolayer and an internal elastic lamina), the media (mainly composed of several layers of smooth muscle cells and elastic fibers), and the adventitia (containing mostly immune cells and fibroblasts). During normal aging, atheroma plaques form predominantly at arterial sites susceptible to atherogenesis due to disturbed blood flow and typically grow eccentrically. As the lesion grows, smooth muscle cells from the media migrate into the intima, proliferate, and produce collagen to form a fibrous cap that protects the plaque from disruption. During progerin-induced aging, smooth muscle cells in the arterial media die, triggering plaque formation over almost the entire intimal surface. These lesions typically contain few smooth muscle cells and display signs of instability, including the presence of erythrocytes and a thin or absent fibrous cap. Vulnerable plaques in HGPS are prone to disruption, thus facilitating thrombus formation and myocardial infarction or stroke. Atherosclerosis in progeric mice is also accompanied by extracellular matrix accumulation in the media, elastin structure alterations, and prominent adventitial thickening.

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
