# Peer review of "Premature Vascular Aging with Features of Plaque Vulnerability in an Atheroprone Mouse Model of Hutchinson–Gilford Progeria Syndrome with Ldlr Deficiency"

_cells, 2020, doi:10.3390/cells9102252_

Round 1
Reviewer 1 Report
The submitted manuscript reports a very interesting analysis of vascular defects associated with the premature ageing syndrome Hutchinson-Gilford Progeria (HGPS). Interestingly, the authors perform a comparison between the LdlR-/- G609G/G609G Lmna mouse model of progeria with defects in lipid profile and the Apoe-/- G609G/G609G Lmna model. Both models, according to the authors and supported by data shown in this manuscript, develop atherosclerotic plaques and altered artery wall structure. The two mouse models and their comparison have been designed in order to avoid bias caused by the different lipid profiles of the two atherosclerosis prone mice.
The main findings of the submitted manuscript are that both models feature smooth muscle cell loss, lack of smooth muscle cell migration to produce collagen caps on atherosclerotic plaques, vulnerable atherosclerotic plaques containing erithrocytes. The latter finding is relevant to the occurrence of strokes, one of the most life-threatening events in HGPS patients.
A few information could be added in this very nice paper:
- is aortic valve calcification, a life-threatening disorder in HGPS, observed in any of the progeroid mouse models here described?
- do the authors find any relevant difference in life-span between atherosclerosis prone mice and the original Lmna G609G mouse model?
- does the LdlR-/- G609G/G609G Lmna mouse model feature the same fat loss (lipodystrophy) pattern as the G609G/G609G Lmna mouse?
- Could the authors speculate that an altered lipid profile might elicit different effects as for atherosclerosis induction in aged versus younger individuals?
- for people not familiar with the G609G/G609G Lmna mouse model, the recently published description by Zaghini et al (2020) could be cited.
Reviewer 2 Report
The study by Nevado and co-authors "Premature vascular aging with features of plaque vulnerability in an atheroprone mouse model of Hutchinson-Gilford progeria syndrome with Ldlr deficiency" aimed at generation of mouse model of Hutchinson-Gilford Progeria Syndrome (HGPS). This is an important step towards more complex study of HGPS mechanisms and early stages of the disease development. Since mere G609G mutation (even homozygous) of LMNA in mice does not fully recreate human HGPS phenotype, especially the vascular components, which are considered the main cause of early death of HGPS patients, the authors decided to introduce LMNA G609G mutation into Ldlr-/- mice, a well known model of atherosclerosis. To further stimulate atherosclerosis development, animals were fed high fat diet, and organismal ageing and atherogenic morphological changes in aorta were studied. The authors claim that the main impact of LmnaG609G/G609G on atherosclerosis progression is the depletion of smooth muscle cells in arterial media and hence plaque vulnerability causing thrombosis. It is still not clear to me whether combination of high fat diet with Ldlr deficiency and homozygous Lmna G609G mutation correctly recapitulates atherogenic effects of HGPS in humans and these mice can be considered a good experimental model. Moreover, in this study no cellular mechanisms involved in accelerated aging were investigated directly, hence I believe the manuscript under consideration is somewhat beyond the scope of the Cells journal, but definitely contains an important new information to be considered for publication in a more specialized journal, for example MDPI's "Journal of Cardiovascular development and Disease".
As minor points, image quantitation procedures are not described in sufficient detail in Materials and Methods, and some typos require correction.
Round 2
Reviewer 2 Report
In its current version the manuscript shows considerable improvements, most of my points have been properly addressed or, at least, authors' explanations are acceptable. Provided the manuscript is intended to be published in a special issue on the therapy of laminopaties, I retract my criticism concerning the relevance of the study to the scope of the Cells journal. The manuscript can be accepted for publication.